# An Assessment of Administration Route on MSC-sEV Therapeutic Efficacy

**DOI:** 10.3390/biom14060622

**Published:** 2024-05-24

**Authors:** Bin Zhang, Ruenn Chai Lai, Wei Kian Sim, Thong Teck Tan, Sai Kiang Lim

**Affiliations:** 1Paracrine Therapeutics Pte. Ltd., 1 Tai Seng Ave, #02-04 Tai Seng Exchange, Singapore 536464, Singapore; bin.zhang@paracrinetherapeutics.com (B.Z.); ruennchai.lai@paracrinetherapeutics.com (R.C.L.); eugene.sim@paracrinetherapeutics.com (W.K.S.); thongteck.tan@paracrinetherapeutics.com (T.T.T.); 2Department of Surgery, YLL School of Medicine, National University Singapore (NUS), 5 Lower Kent Ridge Road, Singapore 119074, Singapore

**Keywords:** mesenchymal stem/stromal cell (MSC), small extracellular vesicles (sEVs), bleomycin (BLM)-induced skin scleroderma (SSc), fibrosis, collagen, immunomodulation

## Abstract

Mesenchymal stem/stromal cell-derived small extracellular vesicles (MSC-sEVs) are promising therapeutic agents. In this study, we investigated how the administration route of MSC-sEVs affects their therapeutic efficacy in a mouse model of bleomycin (BLM)-induced skin scleroderma (SSc). We evaluated the impact of topical (TOP), subcutaneous (SC), and intraperitoneal (IP) administration of MSC-sEVs on dermal fibrosis, collagen density, and thickness. All three routes of administration significantly reduced BLM-induced fibrosis in the skin, as determined by Masson’s Trichrome staining. However, only TOP administration reduced BLM-induced dermal collagen density, with no effect on dermal thickness observed for all administration routes. Moreover, SC, but not TOP or IP administration, increased anti-inflammatory profibrotic CD163^+^ M2 macrophages. These findings indicate that the administration route influences the therapeutic efficacy of MSC-sEVs in alleviating dermal fibrosis, with TOP administration being the most effective, and this efficacy is not mediated by M2 macrophages. Since both TOP and SC administration target the skin, the difference in their efficacy likely stems from variations in MSC-sEV delivery in the skin. Fluorescence-labelled TOP, but not SC MSC-sEVs when applied to skin explant cultures, localized in the stratum corneum. Hence, the superior efficacy of TOP over SC MSC-sEVs could be attributed to this localization. A comparison of the proteomes of stratum corneum and MSC-sEVs revealed the presence of >100 common proteins. Most of these proteins, such as filaggrin, were known to be crucial for maintaining skin barrier function against irritants and toxins, thereby mitigating inflammation-induced fibrosis. Therefore, the superior efficacy of TOP MSC-sEVs over SC and IP MSC-sEVs against SSc is mediated by the delivery of proteins to the stratum corneum to reinforce the skin barrier.

## 1. Introduction

The therapeutic potency of secretion by native mesenchymal stromal/stem cells (MSCs) was first reported in 2008, and this potency was subsequently attributed to extracellular vesicles (EVs), namely the 80–1000 nm microvesicles, by Cammussi and his group [1], and 100–130 nm MSC-sEVs by our group [2]. In 2017, it was reported that the smaller microvesicles with an average diameter of ~160 nm, but not the larger average diameter of ~215 nm, in the population of 80–1000 nm microvesicles were therapeutic [3]. It is now generally acknowledged that 50–200 nm small EVs or sEVs are the active MSC extracellular vesicles [4], and they mediate the activity of their parental MSCs [1,5,6]. MSCs used for the production of MSC-sEVs in this study are cells of an immortalised monoclonal cell line established and described in 2011 [7,8]. The EVs from these cells have been characterised extensively and intensively since then. These characterisations include proteomic analysis [9,10], deep RNA sequencing [10], surface protein analysis [11], and lipid analysis [12]. The proteomic data were deposited in Exocarta and Vesiclepedia.

Like most drugs, the therapeutic efficacy of MSC-sEVs is likely to be dependent on the route or more specifically, the site of administration, to facilitate the delivery of sEVs to target cells or tissues. Although many studies have reported that EVs exhibit homing or tropism activity—such as EVs from lung-, liver-, and brain-tropic tumour cells fusing preferentially with resident cells in their respective tissues [13]—these findings are being re-evaluated. This re-evaluation is due to the use of lipophilic dyes in studies, which are known to generate labelling artifacts [14,15,16]. As MSCs are ubiquitously present in many tissues, MSC-sEVs are not likely to have specific homing or tropism activity. Recently, it was reported that different routes of administration, namely intravenous and intranasal delivery of EVs from human embryonic kidney (HEK) 293 cells in either mouse or monkey, exhibited different biodistribution [17]. Interestingly, no tropism i.e., homing to the kidney was observed. The different biodistribution of EVs when administered through different routes suggests that the therapeutic efficacy may also be different.

To test this hypothesis, we determined the relative therapeutic efficacy of MSC-sEVs on a mouse model of bleomycin (BLM)-induced skin scleroderma (SSc) using different administrative routes. Scleroderma is an autoimmune disease that causes inflammation, and fibrosis or excessive deposits of collagen (https://www.niams.nih.gov/health-topics/scleroderma, accessed on 30 March 2024) and SSc is a localized scleroderma that affects the skin and the structures directly under the skin. The pathology of scleroderma and the therapeutic efficacy of MSCs and MSC-EVs for this condition have been widely reported, and the current state of research has been extensively covered in a recent review [18]. The rationale for using this model is that we could potentially use three different routes of administration, namely topical (TOP), subcutaneous (SC), and intraperitoneal (IP) to deliver MSC-sEVs. We examined the effects of the administration on dermal thickness, fibrosis, and collagen deposits in the skin, and on M2 macrophages present in the skin.

## 2. Materials and Methods

### 2.1. Culture of MSCs and Preparation of MSC-sEVs

Immortalized E1-MYC 16.3 human ESC-derived MSCs were cultured in DMEM with 10% foetal calf serum as previously described [7]. For MSC-sEV preparation, the conditioned medium was prepared by growing 80% confluent cells in a chemically defined medium for three days as previously described [2,19,20]. The defined medium was prepared as follows: 480 mL DMEM (Cat#31053, Thermo Fisher, Waltham, MA, USA), 5 mL NEAA (Cat#11140-050, Thermo Fisher, Waltham, MA, USA), 5 mL glutamine (Cat#25030-081, Thermo Fisher, Waltham, MA, USA), 5 mL sodium pyruvate (Cat#11360, Thermo Fisher, Waltham, MA, USA), 5 mL ITS-X (Cat#51500-056, Thermo Fisher, Waltham, MA, USA), 0.5 mL 2-ME (Cat#21985-02, Thermo Fisher, Waltham, MA, USA). This was supplemented with 0.1 mL bFGF (0.5 ng/μL 0.2%BSA in PBS (+) and 0.005 mL PDGF (100 ng/μL PBS (+)). These latter components were obtained as follows: bovine serum albumin or BSA (Cat#A9647, Sigma-Aldrich, St. Louis, MO, USA), PDGF (100-00 AB CYTOLAB, Karlsruhe, Germany), bFGF (Cat#13256-029, Thermo Fisher, Waltham, MA, USA), and PBS(+) (Cat#14040-133, Thermo Fisher, Waltham, MA, USA). The conditioned medium (CM) was size fractionated via tangential flow filtration and then concentrated 50× using a membrane with a molecular weight cut-off (MWCO) of 100 kDa (Sartorius, Gottingen, Germany). The MSC-sEV preparation was assayed for protein concentration using a Coomassie Plus (Bradford, UK) Assay Kit (Thermo Fisher, Waltham, MA, USA). Only batches of sEV determined via nanoparticle tracking analysis on a ZetaView instrument (Particle Matrix GmbH, Dusseldorf, Germany) to have 1.46 × 10^11^ ± 2.43 × 10^10^ particles per ug protein and a particle modal size of 138.62 ± 4.45 nm using the parameters (sensitivity = 90, shutter = 70, frame rate = 30, min brightness = 25, min area = 5, max area = 1000) were used for this study. In addition, preparations were required to express CD81 and CD73 as determined by western or ELISA. The EV preparations were filtered with a 0.22 μm filter (Merck Millipore, Billerica, MA, USA) and stored in −80 °C freezer.

### 2.2. Fluorescence Labelling of MSC-sEVs

MSC-sEVs were labelled with Alexa Fluor 488 amine-reactive probe (Cat#A30005, Thermo Fisher Scientific) according to the manufacturer’s protocol. Then, 1 mg of MSC-sEVs in 0.8 mL PBS or 0.8 mL PBS were incubated with 1 mg of the Alexa Fluor 488 probe in a final volume of 1 mL of 0.1 M sodium bicarbonate buffer for 1 h with gentle shaking and protected from light. Excess unreacted probes were removed by passing the two mixtures through Bio-Gel P30 gel columns (Cat#7326231, Bio-Rad Laboratories, Hercules, CA, USA). The respective filtrates representing Alexa-Fluor-488-labelled EVs and Alexa-Fluor-488-labelled PBS were sterile filtered through 0.22 μm filters.

### 2.3. Formulation of Oil-in-Water Emulsion of MSC-sEVs

The oil-in-water emulsion of MSC-sEVs was composed of ingredients listed in Table 1. For topical application of MSC-sEVs in the BLM-induced SSc study or skin penetrance study, the emulsion syringe set of MSC-sEVs, with or without fluorescence-labelled with Alexa Fluor 488 as described above, were required to be kept in room temperature. Just before use, the mixture was emulsified by placing the mixture in a two-syringe setup connected by a tube and mixed by alternately pushing the syringe plungers 20 times so that content in one syringe was pushed into the other in an alternate fashion. In the SSc study, each syringe set contained 600 µL of emulsion after being mixed homogeneously, and 50 µL dosage of emulsion was applied to the back skin per mouse. In the skin penetrance study, cream containing only Alexa-Fluor-488-labelled PBS were henceforth referred to as control cream. Control cream carries no EVs and acts as a vehicle control. This is to determine the baseline fluorescence background of residual Alexa Fluor 488 amine-reactive probe and the oil-in-water emulsion.

### 2.4. BLM-Induced SSc Mouse Model

This study was performed by SMC Laboratories, Inc. (Tokyo, Japan) 2-16-1 Minami-Kamata Ota-City Tokyo 144-0035 Japan under IACUC no: A012. Six-week-old female C57BL/6J mice were obtained from Japan SLC, Inc. (Hamamatsu, Japan). These mice were housed and fed with a normal diet (CE-2; CLEA Japan, Shizuoka, Japan) under controlled conditions. Mice were identified by ear punch and randomized into 5 groups of 10 mice based on their body weight on the day before the start of bleomycin (BLM) hydrochloride (Nippon Kayaku, Tokyo, Japan) administration. Five normal mice served as the normal control group. At day 0, the back skins of the mice were shaved, and 50 mice were induced to develop SSc by subcutaneous administration of BLM in saline at a dose of 50 μg/mouse, in a volume of 50 µL every other day (Day 0 to 26) [21,22,23]. Five control mice were subcutaneously administered saline, instead of the BLM, in a volume of 50 µL every other day (Day 0 to 26). The injection sites were located at the corners of a 1.5 × 1.5 cm square shaved area. The MSC-sEVs were prepared as described above, and the imatinib (purchased from Novartis Pharma K.K., Tokyo, Japan) was suspended in pure water prior to administration as a positive control. MSC-sEVs were administered intraperitoneally, subcutaneously, or applied topically at a dose of 5 μg/50 μL/mouse daily from Day 0 to 27. Imatinib was administered intraperitoneally at a dose of 50 mg/5mL/kg daily from Day 0 to 27 as a positive control. For treatment control, the PBS was intraperitoneally administered at a volume of 50 μL/mouse daily from Day 0 to 27. The viability, clinical signs and behaviour were monitored daily. Body weight was recorded daily before the treatment. Mice were observed for significant clinical signs of toxicity, moribundity, and mortality approximately 60 min after each administration. The animals were sacrificed at Day 28 by exsanguination through abdominal aorta under three types of mixed anaesthetic agents (medetomidine, midazolam, butorphanol). For skin samples, the pre-shaved back skin was collected, snap frozen in liquid nitrogen, and stored at −80 °C for analysis. For histological analysis, the pre-shaved back skin was fixed in 10% neutral buffered formalin for 24 h. After fixation, these specimens were processed to paraffin embedding for HE and MT staining.

### 2.5. Skin Collagen and Histological Analysis

Skin collagen content or density was quantified by Sircol soluble collagen assay kit (Biocolor Ltd., Carrickfergus, UK). For HE staining, sections were cut from paraffin blocks of skin tissue prefixed in 10% neutral buffered formalin and stained with Lillie-Mayer’s Hematoxylin (Muto Pure Chemicals Co., Ltd., Tokyo, Japan) and eosin solution (FUJIFILM Wako Pure Chemical Corporation, Tokyo, Japan). For quantitative analysis of dermal thickness, bright field images of the HE-stained section were captured using a digital camera (DFC295; Leica, Wetzlar, Germany) at 100-fold magnification, and the dermal thickness in 5 fields/section was measured using ImageJ software (1.52v, National Institute of Health, Bethesda, MD, USA). For Masson’s Trichrome (MT) staining, sections were cut from paraffin blocks of skin tissue prefixed in 10% neutral buffered formalin and stained with Masson’s Trichrome Staining Kit (Sigma, Burlington, MA, USA) according to the manufacturer’s instructions. For quantitative analysis of the fibrosis area, bright field images of Masson’s Trichrome-stained sections were captured using a digital camera (DFC295) at 100-fold magnification, and the fibrosis area in 5 fields/section was measured using ImageJ software.

### 2.6. Immunohistochemistry Staining for CD163

As a marker of M2 macrophages, CD163 was assessed [24] by standard immunohistochemistry (IHC). The SSc skin sections at 4 μm cut from paraffin blocks were routinely dewaxed and rehydrated. Endogenous peroxidase activity was blocked using 0.3% H_2_O_2_ for 5 min. The slides were then incubated with antigen retrieval reagent (RM102-H, LSI Medience Corporation, Tokyo, Japan) for 10 min at 121 °C. The sections were incubated with anti-CD163 antibodies (clone no. EPR19518, Rabbit monoclonal, Abcam, Cambridge, UK) at 4 °C overnight. After incubation with the secondary antibody (VECTASTAIN^®^ Elite ABC-HRP Kit, Vector laboratories, Inc., Newark, CA, USA), enzyme-substrate reactions were performed using three 3′-diaminobenzidine/H_2_O_2_ solution (Nichirei Bioscience Inc., Tokyo, Japan). For quantitative analysis of CD163-positive areas, bright field images of CD163-immunostained sections were captured using a digital camera (DFC295) at 200-fold magnification, and the positive areas in 5 fields/section were measured using ImageJ software (1.52v, National Institute of Health, Bethesda, MD, USA).

### 2.7. Human Skin Penetrance Assay

The penetrance study was performed by DeNova Sciences Pte Ltd. Full thickness discarded human skin from an abdominoplasty procedure was obtained with informed patient consent and proper DSRB approval (NHG DSRB 2016/00525, 8 September 2016 and AC-2017-3030) from Liberty Equality Fratered, Republic Francaise, Ministry of Higher Education and Research. The hypodermis/fats were removed, and only the intact skin consisting of dermis and epidermis was kept. After that, the skin was soaked in 70% ethanol for 2 s to remove contamination, rinsed in 1× PBS for 3 min, and then placed in DeNova decontamination medium for 2 h in a 37 °C CO_2_ incubator. Thereafter, the skin explants were cut into 0.5 × 2.5 cm strips. The strips had an average thickness of 0.45 ± 0.05 cm and surface area of 1.0 cm^2^. The strips were placed with the apical epithelial surface up in the Transwell permeable supports of a 6-well plate with 1 mL of DeNova skin explant medium maintained at the air-medium interface. For topical application, 20 μL of cream, including Alexa-Fluor-488-labelled MSC-sEVs (EV cream) or Alexa-Fluor-488-labelled PBS (Control cream) was directly applied to the apical surface of the strips. For subcutaneous injection, 20 μL of Alexa-Fluor-488-labelled MSC-sEVs or Alexa-Fluor-488-labelled PBS was directly injected. The strips were incubated for 12 or 24 h in a 37 °C incubator. At the end of each incubation period, the culture media were collected, frozen, and stored. The skin explants were gently washed with PBS, blotted dry, placed in OCT quick-freeze medium compound, and then snap-frozen in liquid nitrogen. The 3 μm cryotissue sections were cut and mounted on SuperFrost Plus slides. The sections were then immunostained with a primary antibody against Laminin 5 (Merck-Millipore, Rahway, NJ, USA) (1:200) and a secondary antibody (Alexa fluor A594, Life Technologies, Carlsbad, CA, USA) (1:1000). After DAPI nuclear staining, the sections were viewed using a Carl Zeiss Axio Scan Z.1 microscope (Carl Zeiss, Jena, Germany) and Zen Blue software (ZEN 2.5) with constant exposure and gain.

### 2.8. Statistical Analysis

Statistical analyses were performed using the Bonferroni Multiple Comparison Test on GraphPad Prism 6 (GraphPad Software Inc., San Diego, CA, USA). The *p* values < 0.05 were considered statistically significant. The average value and standard deviation of each group were calculated by the individual animal in the group. A trend or tendency was assumed when a one-tailed *t*-test returned *p* values < 0.1. Results were expressed as mean ± SD.

## 3. Results

### 3.1. Effects of Different Routes of MSC-sEV Administration on Dermal Thickness, Area of Fibrosis, and Collagen Density in SSc Mouse Model

To assess the effect of MSC-sEVs on SSc mice, we used BLM-induced skin fibrosis in mice to model SSc [21]. MSC-sEVs were administered to this mouse model by IP, SC, or TOP applications. The positive control was the imatinib-treated group at a dose of 50 mg/5 mL/kg. MSC-sEVs were administered intraperitoneally, subcutaneously, or topically at a dose of 5 μg/50 μL/mouse. The treatment control group was the IP administered SSc mice with 50 μL saline/mouse. The animals in all treatment groups, i.e., MSC-sEV administration, positive control, and treatment control groups, had statistically higher dermal and lower adipose layer thickness than normal mice (Figure 1A–C). The difference among all these groups was not statistically significant in dermal thickness. Since it is a surrogate marker of disease severity [25], the similarity in dermal thickness indicated that all treatment and control groups had similar disease severity. In contrast, all modes of MSC-sEV administration had statistically significant reduction in the fibrotic area as measured by Masson’s Trichrome staining when compared to the positive and treatment control groups (Figure 1D,E). Of the three groups of MSC-sEV-treated animals, only TOP animals exhibited a statistically significant reduction in collagen content or density compared to the treatment control groups (Figure 1F). The collagen density in TOP MSC-sEV mice was statistically similar to that of normal mice (13.4 ± 5.7 vs. 21.6 ± 8.4 mg/mm^2^ skin, *p* = 0.041). Therefore, TOP is the most efficacious mode of administration to reduce skin fibrosis.

### 3.2. Polarization of MSC-sEVs on Anti-Inflammatory M2 Macrophages

We have previously reported that our MSC-sEVs promote M2 macrophage polarization [26,27,28]. As M2 macrophages have been implicated as pro-fibrotic [29], we determined the infiltration of anti-inflammatory CD163^+^ M2 macrophages in the BLM-induced skin lesions after administrations of MSC-sEVs. We observed that relative to the treatment control group, the SC, but not the IP or TOP treatment group, had a statistically significant higher infiltration of CD163^+^ cells (5.4 ± 2.0 vs. 1.0 ± 0.2, *p* = 0.010) (Figure 2). As IP MSC-sEVs did not reduce collagen density while TOP MSC-sEVs did when neither route of administration increased M2 macrophage, the level of M2 macrophage was not associated with the reduced collagen density in TOP mice. 

### 3.3. Spatiotemporal Distribution of MSC-sEVs in Skin

While both SC and TOP administrations target the skin [30], TOP was more effective than SC in reducing collagen density. Therefore, the delivery of MSC-sEVs to the skin via these administration routes is likely to be more nuanced. Consistent with our previous report [30], it was observed that when TOP fluorescence-labelled MSC-EVs were applied to skin explant cultures, the fluorescence predominantly localized to the stratum corneum for up to 24 h, with minimal fluorescence transitioning to the underlying nucleated stratum granulosum (Figure 3A). In contrast, we observed here that there was no distinct localization of fluorescence in the stratum corneum for the SC MSC-sEVs (Figure 3B), demonstrating a significant difference in the site of MSC-sEV delivery between TOP and SC administration and implying that this different localization is the reason for the discrepancy in efficacy between the two modes of administration. To rationalize the relationship between the locale of the MSC-sEVs after administration and the therapeutic efficacy on SSc, we compared the proteomes of the stratum corneum [31] and our MSC-sEVs [10]. There were 109 common proteins (Table 2). Many were structural proteins such as keratins that are important for maintaining skin barrier integrity and function. One particularly notable protein is filaggrin. Filaggrins are crucial in enhancing skin barrier function by promoting the adhesion and mechanical strength of corneocytes in the stratum corneum [32,33]. Therefore, unlike SC administration, TOP administration of MSC-sEVs delivers critical structural proteins to strengthen the skin barrier and prevent infiltration of irritants to induce inflammation and inflammation-induced fibrosis.

## 4. Discussion

To date, numerous studies have delved into the biodistribution and pharmacokinetics of EVs [34,35,36,37,38]. However, a significant gap remains in our understanding regarding how the route of EV administration influences its biodistribution. This gap persists due to the limited number of studies directly comparing administration routes. For instance, in one study, EVs were administered to mice via intravenous, subcutaneous, and intraperitoneal routes [39]. Interestingly, compared to intravenous administration, both subcutaneous and intraperitoneal routes led to decreased EV uptake in the liver and spleen but increased uptake in the gastrointestinal tract and pancreas. Conversely, another study found that intranasal administration of EVs exhibited enhanced brain targeting compared to intravenous administration [40].

In summary, our study illustrates the influence of the administration route on the therapeutic efficacy of MSC-sEVs. Through experiments on mice with BLM-induced SSc, we found that TOP administration was most effective in reducing both fibrotic area and collagen density. Surprisingly, SC administration, despite also targeting the affected skin, was not as effective and only matched systemic IP administration in reducing fibrotic area, not collagen density. Although SC administration increased M2 macrophage levels in the skin, this did not translate to improvements in fibrotic area or collagen density compared to other treatment groups.

These observations led us to hypothesize s that the superior outcome of TOP over SC administration is due to differing biodistribution of MSC-sEVs in the skin. Indeed, we observed that TOP-administered EVs were primarily concentrated in the stratum corneum, unlike SC-administered EVs. Comparative analysis of MSC-sEVs and stratum corneum proteome revealed 109 proteins common to both, mostly structural proteins crucial for skin barrier function. This aligns with previous studies linking reduced concentration of structural proteins like keratins and filaggrin to compromised skin barrier function [41,42,43,44]. Filaggrin is one of the key structural proteins essential for the adhesion and mechanical functions of corneocytes in maintaining skin barrier function [45]. We noted that TOP application of filaggrin monomers could restore the skin barrier in filaggrin-deficient flaky tail (ft/ft) mice [46], highlighting the potential of stratum corneum-like proteins in TOP MSC-sEVs to augment the skin barrier.

Delivery of filaggrin to the stratum corneum by TOP MSC-sEVs is pertinent to alleviating SSc. In SSc skin, filaggrin is mainly found in the lower epidermal layer instead of the stratum corneum, depriving the SSc stratum corneum of the critical filaggrin to maintain the adhesion and mechanical strength of corneocytes for skin barrier integrity [47]. A compromised skin barrier will lead to increased permeability to irritants, inflammation, and subsequent fibrosis. Hence, TOP, unlike SC or IP, MSC-sEV administration could deliver filaggrin to compensate for its deficiency in the SSc stratum corneum and restore skin barrier function, offering a promising therapeutic avenue to reduce skin inflammation and inflammation-induced fibrosis in SSc.

## 5. Conclusions

In conclusion, our study highlights the greater efficacy of TOP administration of MSC-sEVs in treating SSc compared to SC and IP routes. TOP administration facilitates the delivery of essential stratum corneum proteins, such as filaggrin, found in MSC-sEVs, thereby enhancing the skin barrier function. This underscores the importance of administration routes in EV-based therapies.

## Figures and Tables

**Figure 1 biomolecules-14-00622-f001:**
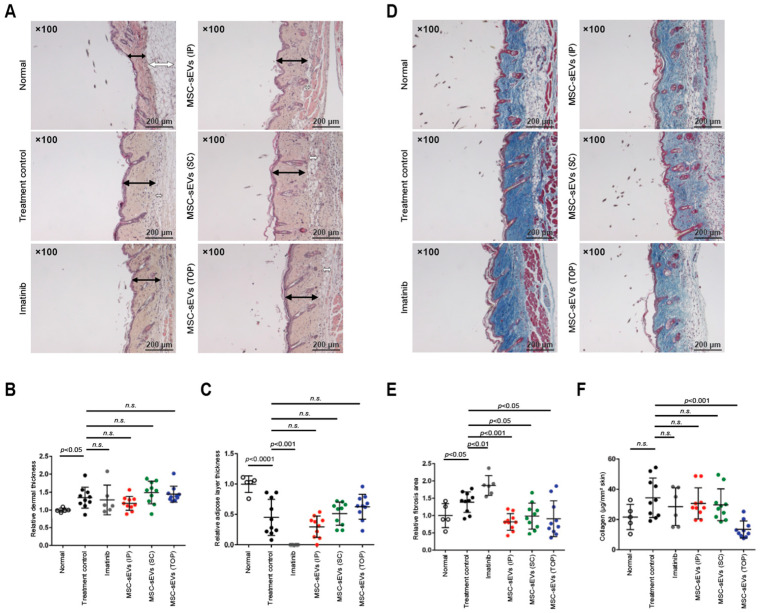
Evaluation of anti-fibrosis mediated by MSC-sEVs in SSc mouse model. (**A**) Representative photomicrographs of HE-stained skin sections are shown. The sections were cut from paraffin blocks of skin tissue prefixed in 10% neutral buffered formalin and stained with Lillie-Mayer’s Hematoxylin and eosin solution. Black/white arrows represent dermal/adipose layer thickness. (**B**,**C**) The quantitative analysis of dermal/adipose layer thickness. The bright field images of HE-stained section were captured using a digital camera at 100-fold magnification, and the dermal/adipose layer thickness in 5 fields/section were measured using ImageJ software. (**D**) Representative photomicrographs of Masson’s Trichrome-stained skin sections are shown. The sections were cut from paraffin blocks of skin tissue prefixed in 10% neutral buffered formalin and stained with Masson’s Trichrome staining kit according to the manufacturer’s instructions to visualize collagen deposition. (**E**) The quantitative analysis of fibrosis area. The bright field images of Masson’s Trichrome-stained sections were captured using a digital camera at 100-fold magnification, and the fibrosis areas in 5 fields/section were measured using ImageJ software. (**F**) Skin collagen content is shown. The quantification of skin collagen content was measured by Sircol soluble collagen assay kit according to the manufacturer’s instructions. The “*n.s.*” represents “not significant”.

**Figure 2 biomolecules-14-00622-f002:**
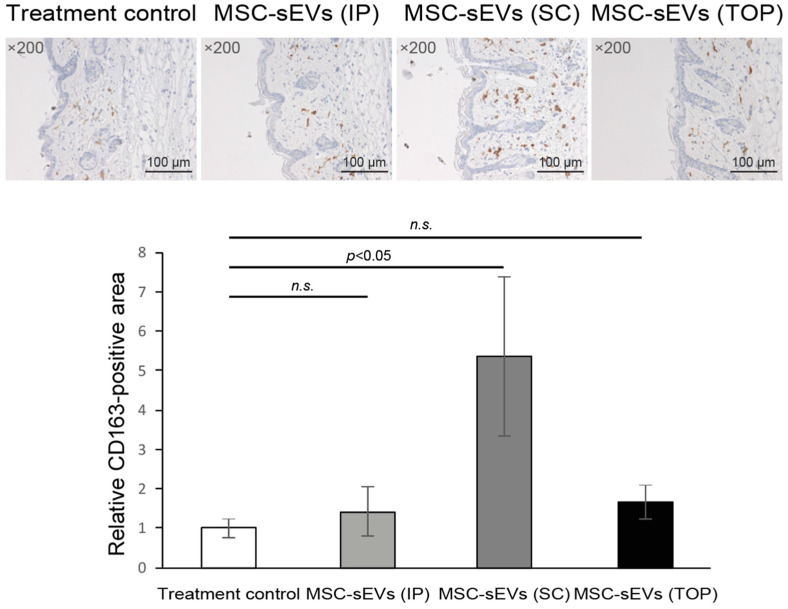
Effects of MSC-sEVs on M2 macrophage infiltration. Representative photomicrographs of CD163-immunohistochemistry-stained skin (**upper panel**) sections of SSc mice are shown. For quantitative analysis of CD163-positive areas, bright field images of CD163-immunostained sections were captured using a digital camera at 200-fold magnification, and the CD163-positive areas in 5 fields/section were measured using ImageJ software in skin sections (**lower panel**). Data represent mean ± SD compared to vehicle control. The “*n.s.*” represents “not significant”.

**Figure 3 biomolecules-14-00622-f003:**
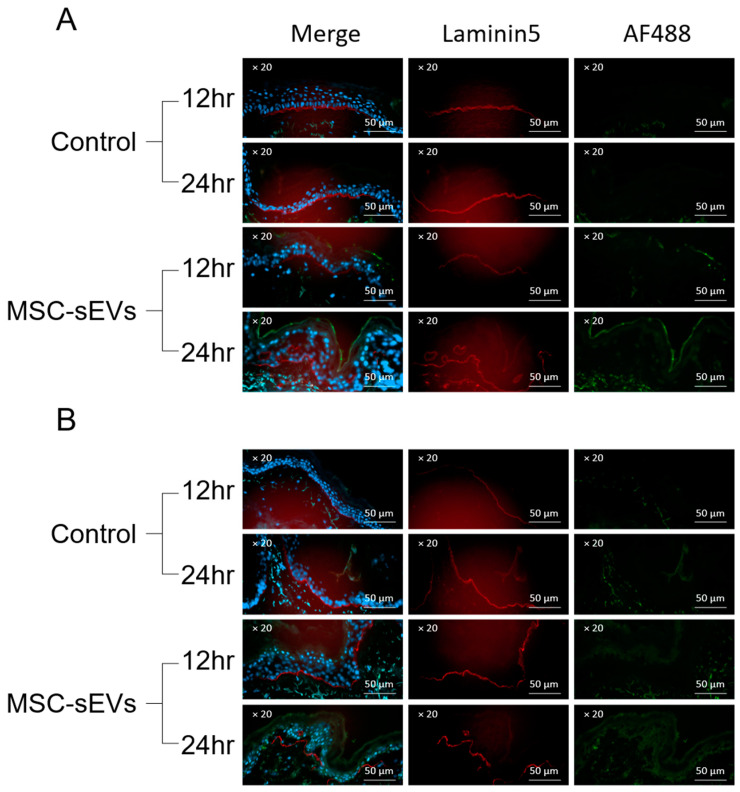
Spatiotemporal distribution of topically applied versus subcutaneously injected MSC-sEVs in intact skin. (**A**) Human skin explants were topically treated with Control cream (including Alexa-Fluor-488-labelled PBS) or EV cream (including Alexa-Fluor-488-labelled MSC-sEVs) for 12 and 24 h or (**B**) subcutaneously injected with Alexa-Fluor-488-labelled PBS Control or Alexa-Fluor-488-labelled MSC-sEVs for 12 or 24 h, washed, frozen in OCT medium, and sectioned. Fluorescence imaging of sections was counterstained with DAPI (blue) to show nuclei and anti-laminin 5 (red) to define the epidermal–dermal junction.

**Table 1 biomolecules-14-00622-t001:** Formulation of oil-in-water emulsion of MSC-sEVs.

	Volume or Weight	Final Concentration
Olive oil	200 µL	20% *v*/*v*
MSC-sEVs	100 or 400 µg	100 or 400 µg/mL
100% Seppic plus 400 *	40 mg	4% *w*/*v*
PBS	400 µL	40% *v*/*v*
water	400 µL	40% *v*/*v*

* Seppic 400 is a commercial emulsifying agent consisting of Polyacrylate-13, Polyisobutene & Polysorbate 20 sold by Seppic, https://www.seppic.com (accessed on 21 March 2019).

**Table 2 biomolecules-14-00622-t002:** The analysis of overlapping proteins between stratum corneum and MSC-sEVs.

ACTB	AHCY	AHNAK	ALDH16A1	ALDH7A1
ALDOA	ANXA1	ANXA2	ANXA4	ANXA7
ARF6	BCAP31	BLMH	CAPN1	CPNE3
CTSA	DCXR	DNM1L	DSP	EEF1A1
EEF2	EIF6	ENO1	FLG2	GBA
GDPD3	GNB2	HIST1H4A	HNRNPA2B1	HRNR
HSD17B4	HSPA1A	HSPA5	HSPA6	HSPB1
IDE	IDH1	IGHG1	IL37	JUP
KRT1	KRT10	KRT13	KRT14	KRT15
KRT16	KRT17	KRT19	KRT2	KRT3
KRT32	KRT36	KRT4	KRT77	KRT78
KRT81	KRT82	KRT85	KRT9	LDHA
LGALS7	LYPLA1	MDH2	ME1	NPEPPS
NPM1	NUDT5	PARK7	PLSCR3	PNP
PRCP	PRDX1	PRDX2	PRDX4	PSMA1
PSMA2	PSMA3	PSMA4	PSMA5	PSMA7
PSMB1	PSMB2	PSMB3	PSMB5	PSMB6
PSMB7	RAB10	RAB35	RAB5A	RAB5B
RALA	S100A10	S100A11	S100A9	SCARB2
SCPEP1	SERPINB12	SERPINB6	SPTBN2	SYPL1
TALDO1	TKT	TMED10	TPI1	TPP1
TROVE2	TXN	VAT1	WDR77	

## Data Availability

The data presented in this study are all available in this article.

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
