# Peer review of "An Assessment of Administration Route on MSC-sEV Therapeutic Efficacy"

_biomolecules, 2024, doi:10.3390/biom14060622_

Round 1
Reviewer 1 Report
Comments and Suggestions for Authors
It's an interesting report to show us the TOP administration of hESC-MSC EVs to treat the mouse SSC model. I have some concerns that should be addressed in the manuscript.
Major points:
1. This article should identify the MSCs and hESC-MSC-EVs. Especially the characters of the hESC-MSC-EVs.
2. The BLM-induced SSc model has atrophy of adipose tissue. Could you provide the quantification of the adipose layer of the treatments?
3. All the sections of your results show the hair missing from the root of the hair follicles. You mentioned that the methods included shaving the back skins of the mice. Please provide the detailed protocols.
Minor points:
1. The scale bar of the figures related to sections should be added, not just the multiples.
Author Response
Major points:
- This article should identify the MSCs and hESC-MSC-EVs. Especially the characters of the hESC-MSC-EVs.
Response 1: We would like to clarify that the MSCs used for the production of MSC-sEVs in this manuscript are cells of an immortalised monoclonal cell line established and described in 2011 [1, 2]. The EVs from these cells have been characterised extensively and intensively since then. These characterisation include proteomic analysis [3, 4], deep RNA sequencing [4], surface protein analysis [5] and lipid analysis [6]. The proteomic data were deposited in Exocarta and Vesiclepedia. Given the extensive and intensive characterisation details that have been reported in publicly accessible reports, we chose not to describe the characterisation of the EVs. However, we acknowledge the validity of reviewer’s point, and we will add the above citations in the manuscript to facilitate access to these characterisation data.
- The BLM-induced SSc model has atrophy of adipose tissue. Could you provide the quantification of the adipose layer of the treatments?
Response 2: Thank you for your valuable suggestion. We have quantified the fat layer and accordingly revised Figure 1 and the results text.
- All the sections of your results show the hair missing from the root of the hair follicles. You mentioned that the methods included shaving the back skins of the mice. Please provide the detailed protocols.
Response 3: The hair on the back of mouse neck was shaved (1.5 cm square where BLM was administered) using a hair clipper (Natsume Seisakusho Co., Ltd., Japan) but not pulled out. Perhaps the hairs were stripped off during the thinning of the paraffin block or in a subsequent process.
Minor points:
- The scale bar of the figures related to sections should be added, not just the multiples.
Response 1: In response to the reviewer’s feedback, we have accordingly added the scale bar of the figures.
Reference:
- Chen, T.S., et al., Enabling a robust scalable manufacturing process for therapeutic exosomes through oncogenic immortalization of human ESC-derived MSCs. J Transl Med, 2011. 9: p. 47.
- Lai, R.C., A. Choo, and S.K. Lim, Derivation and characterization of human ESC-derived mesenchymal stem cells. Methods Mol Biol, 2011. 698: p. 141-50.
- Lai, R.C., et al., Proteolytic Potential of the MSC Exosome Proteome: Implications for an Exosome-Mediated Delivery of Therapeutic Proteasome. Int J Proteomics, 2012. 2012: p. 971907.
- Lai, R.C., et al., MSC secretes at least 3 EV types each with a unique permutation of membrane lipid, protein and RNA. Journal of extracellular vesicles, 2016. 5: p. 29828-29828.
- van Balkom, B.W.M., et al., Proteomic Signature of Mesenchymal Stromal Cell-Derived Small Extracellular Vesicles. PROTEOMICS, 2019. 19(1-2): p. 1800163.
- Lai, R.C. and S.K. Lim, Membrane lipids define small extracellular vesicle subtypes secreted by mesenchymal stromal cells. J Lipid Res, 2019. 60(2): p. 318-322.

Reviewer 2 Report
Comments and Suggestions for Authors
Dear Authors,
The manuscript entitled "An assessment of administration route on MSC-sEV therapeutic efficacy" provides information regarding the potential use of MSCs-EVs in a mouse model of skin scleroderma. Major revisions are required to be performed for the submitted manuscript, where below yoy will find my comments.
1) The title of the manuscript needs revision. The title cannot reflect well the main scope of the manuscript. Please include in the title the term of "scleroderma-mouse model"
2) The introduction is very brief and not well-structured. If the authors want to emphasize the action of EVs in skin scleroderma, then should start primary with the disease, followed by the use of the MSCs. In addition, more information regarding the MSCs and the latest updates outlined by the ISCT also should be mentioned.
3) In materials and methods section, the authors indicated the use of E1-MYC 16.3 human ESC-derived MSCs for their performed experiments. However, i cannot understand the reasons why the authors, used this cell-line istead of using primary cells obtained either from the human umbilical cord, bone marrow or adipose tissue. Therefore, the authors, should perform assays to confirm the minimum criteria as outlined by the ISCT-MSC committee for confirming the properties of MSCs.
4) The authors should also include in all experiments MSCs from a different human source such as human umbilical cord, bone marrow or adipose tissue, in order to compare further their results.
5) The conditioned medium used for the isolation of MSCs-EVs contained at least 1% BSA. BSA may also contain EVs, for this reason the authors should perform the same experimental study, without the using of BSA, to compare potential differences in EVs.
6) Is there any particular reason, why the authors used bFGF, PDGF and ITS in the conditioned medium.
7) How many hours the conditioned medium was remained with the MSCs before the isolation of EVs.
8) Please provide detailed information regarding the EVs isolation process.
9) In Figure 1, please provide images with higher magnification. Also use black arrows to show the alterations of skin thickness between the different conditions.
10) Besides the Masson's Trichrome, perform also Sirius Red for better characterization of the collagen fibers alignment.
11) The authors performed experiments regarding only the M2 macrophages. What about M1 macrophages which is considered as a highly inflammatory cell population.
12) In Figure 3, please add original magnification and scale bars of the microscopic images.
13) The authors should perform full proteomic analysis (e.g. mass spectrometry) of the content of MSCs-EVs
14) The discussion is too small and need more similar studies from different groups to be added and discussed thorougly.
Author Response
1) The title of the manuscript needs revision. The title cannot reflect well the main scope of the manuscript. Please include in the title the term of "scleroderma-mouse model"
Response 1: We respectfully disagree with the reviewer’s suggestion to include the term "scleroderma-mouse model" in the title. Our choice of title was to accurately reflect the intent of our research to demonstrate the effects of administration route on the therapeutic outcome of MSC-sEVs, and not the therapeutic effect of MSC-sEVs on scleroderma per se. The scleroderma model being a dermal disease was a convenient disease model to test the three administration routes – IP, TOP and SC. The inclusion of scleroderma in our title will detract the significance of administration route from our study. We hope that you understand our rationale for maintaining the current title.
2) The introduction is very brief and not well-structured. If the authors want to emphasize the action of EVs in skin scleroderma, then should start primary with the disease, followed by the use of the MSCs. In addition, more information regarding the MSCs and the latest updates outlined by the ISCT also should be mentioned.
Response 2: We appreciate reviewer’s suggestion. However, we would like to clarify that our primary focus in this study is on the impact of different administration routes on the therapeutic efficacy of MSC-sEVs, rather than the action of EVs in skin scleroderma. The pathology of scleroderma and the therapeutic efficacy of MSCs and MSC-EVs for this condition have been widely reported. The current state of research has been extensively covered in a recent review [1]. We will include this reference in our manuscript to address the reviewer’s concern about the lack of information on the disease and the use of MSCs and MSC-EVs.
Our manuscript aims to address a significant bottleneck in the development of MSC-EV-based therapeutic applications: the optimization of the administration route. We used scleroderma as a model to assess three possible routes of administration, namely topical, subcutaneous, and intraperitoneal, and to demonstrate the importance of administration methods on the efficacy of MSC EVs in modulating disease pathology.
The description of the MSCs and MSC-EVs in our manuscript has been addressed in response 3 shown below. This includes adherence to recommendations from workshops organized jointly by ISCT and ISEV.
3) In materials and methods section, the authors indicated the use of E1-MYC 16.3 human ESC-derived MSCs for their performed experiments. However, i cannot understand the reasons why the authors, used this cell-line istead of using primary cells obtained either from the human umbilical cord, bone marrow or adipose tissue. Therefore, the authors, should perform assays to confirm the minimum criteria as outlined by the ISCT-MSC committee for confirming the properties of MSCs.
Response 3: It is now widely accepted that the use of immortalized MSCs is critical to the development of robust scalable GMP-compliant manufacture of EVs [2-5]. Our use of immortalized MSCs for EV production is consistent with current best practices. We would like to clarify that the MSCs used for the production of MSC-sEVs in this manuscript are monoclonal cells of an immortalised cell line established and described in 2011 [6, 7]. The EVs from these cells have been characterised extensively and intensively since then. These characterisation include proteomic analysis [8, 9], deep RNA sequencing [9], surface protein analysis [10] and lipid analysis [11]. The proteomic data were deposited in Exocarta and Vesiclepedia.
4) The authors should also include in all experiments MSCs from a different human source such as human umbilical cord, bone marrow or adipose tissue, in order to compare further their results.
Response 4: We previously demonstrated that our MSCs were comparable to those derived from bone marrow and adipose tissues [12], as well as from fetal liver, limb, and kidney [13]. Following immortalization, the cells retained similarity to their unimmortalized counterparts [6, 7]. We determined the proteome of EVs from both our non-immortalized and immortalized MSCs [8, 9], and the resulting data have been deposited in public databases, ExoCarta, and Vesiclepedia. A proteomic comparison of EVs from our MSCs with those derived from placental chorionic villi, bone marrow, umbilical cord, and adipose tissues revealed a common MSC-EV protein signature [10].
5) The conditioned medium used for the isolation of MSCs-EVs contained at least 1% BSA. BSA may also contain EVs, for this reason the authors should perform the same experimental study, without the using of BSA, to compare potential differences in EVs.
Response 5: We are unsure how the reviewer determined that our conditioned medium contained at least 1% BSA and suggested that BSA might harbor EVs. As previously demonstrated, our analysis revealed that, unlike the conditioned medium, the HPLC fractionation profile of the unconditioned medium lacked a peak indicative of EVs carrying CD9 [14]. Consequently, irrespective of the reviewer's suggestion, the unconditioned medium did not contain EVs.
6) Is there any particular reason, why the authors used bFGF, PDGF and ITS in the conditioned medium.
Response 6: We have previously observed that in the absence of serum, our cells could proliferate in the presence of bFGF, PDGF and ITS [7, 12, 15].
7) How many hours the conditioned medium was remained with the MSCs before the isolation of EVs.
Response 7: As describe on line 70, “…. the conditioned medium was prepared by growing 80% confluent cells in a chemically defined medium for three days …”
8) Please provide detailed information regarding the EVs isolation process.
Response 8: As described on line 80 “… The conditioned medium (CM) was size-fractionated by tangential flow filtration and then concentrated 50× using a membrane with a molecular weight cut-off (MWCO) of 100kDa (Sartorius, Gottingen, Germany).”
9) In Figure 1, please provide images with higher magnification. Also use black arrows to show the alterations of skin thickness between the different conditions.
Response 9: Thank you for your valuable feedback. We have revised Figure 1 in accordance with your suggestions.
10) Besides the Masson's Trichrome, perform also Sirius Red for better characterization of the collagen fibers alignment.
Response 10: We did perform a Sirius Red staining assay. The collagen concentration in Fig 1F was measured using Sircol soluble collagen assay kit and this kit used Sirius red dye to determine collagen concentration.
11) The authors performed experiments regarding only the M2 macrophages. What about M1 macrophages which is considered as a highly inflammatory cell population.
Response 11: We have consistently shown that our MSC-sEVs polarised monocytes towards a M2 and not a M1 phenotype [16-19]. Therefore, we did not test for the presence of M1 macrophages.
12) In Figure 3, please add original magnification and scale bars of the microscopic images.
Response 12: In response to your feedback, we have accordingly revised Figure 3 to align with your suggestions.
13) The authors should perform full proteomic analysis (e.g. mass spectrometry) of the content of MSCs-EVs
Response 13: The proteomic analysis of the MSC EVs has been performed and reported [8, 9], and the proteome data have been deposited in public databases, ExoCarta, and Vesiclepedia. Comparison of the proteome of our EVs with those from MSCs derived from placental chorionic villi, bone marrow, umbilical cord, and adipose tissues showed that all the EVs shared a common MSC-EV protein signature [10].
14) The discussion is too small and need more similar studies from different groups to be added and discussed thoroughly.
Response 14: We are grateful for the reviewer's suggestion and have incorporated relevant content into the discussion section accordingly.
Reference:
- Zhang, Y., et al., Research progress on mesenchymal stem cells and their exosomes in systemic sclerosis. Front Pharmacol, 2023. 14: p. 1263839.
- Rani, S., et al., Mesenchymal Stem Cell-derived Extracellular Vesicles: Toward Cell-free Therapeutic Applications. Mol Ther, 2015. 23(5): p. 812-823.
- Lai, R.C., et al., A roadmap from research to clinical testing of mesenchymal stromal cell exosomes in the treatment of psoriasis. Cytotherapy, 2023. 25(8): p. 815-820.
- Mouloud, Y., et al., Immortalization strategies for human mesenchymal stromal cells for large scale production of extracellular vesicles. Cytotherapy, 2020. 22(5, Supplement): p. S53-S54.
- Johnson, J., et al., From Mesenchymal Stromal Cells to Engineered Extracellular Vesicles: A New Therapeutic Paradigm. Frontiers in Cell and Developmental Biology, 2021. 9.
- Chen, T.S., et al., Enabling a robust scalable manufacturing process for therapeutic exosomes through oncogenic immortalization of human ESC-derived MSCs. J Transl Med, 2011. 9: p. 47.
- Lai, R.C., A. Choo, and S.K. Lim, Derivation and characterization of human ESC-derived mesenchymal stem cells. Methods Mol Biol, 2011. 698: p. 141-50.
- Lai, R.C., et al., Proteolytic Potential of the MSC Exosome Proteome: Implications for an Exosome-Mediated Delivery of Therapeutic Proteasome. Int J Proteomics, 2012. 2012: p. 971907.
- Lai, R.C., et al., MSC secretes at least 3 EV types each with a unique permutation of membrane lipid, protein and RNA. Journal of extracellular vesicles, 2016. 5: p. 29828-29828.
- van Balkom, B.W.M., et al., Proteomic Signature of Mesenchymal Stromal Cell-Derived Small Extracellular Vesicles. PROTEOMICS, 2019. 19(1-2): p. 1800163.
- Lai, R.C. and S.K. Lim, Membrane lipids define small extracellular vesicle subtypes secreted by mesenchymal stromal cells. J Lipid Res, 2019. 60(2): p. 318-322.
- Lian, Q., et al., Derivation of clinically compliant MSCs from CD105+, CD24− differentiated human ESCs. Stem cells, 2007. 25(2): p. 425-436.
- Lai, R.C., et al., Derivation and characterization of human fetal MSCs: an alternative cell source for large-scale production of cardioprotective microparticles. J Mol Cell Cardiol, 2010. 48(6): p. 1215-24.
- Lai, R.C., et al., Exosome secreted by MSC reduces myocardial ischemia/reperfusion injury. Stem Cell Res, 2010. 4(3): p. 214-22.
- Sze, S.K., et al., Elucidating the secretion proteome of human embryonic stem cell-derived mesenchymal stem cells. Mol Cell Proteomics, 2007. 6(10): p. 1680-9.
- Zhang, B., et al., Therapeutic Efficacy of Mesenchymal Stem/Stromal Cell Small Extracellular Vesicles in Alleviating Arthritic Progression by Restoring Macrophage Balance. Biomolecules, 2023. 13(10).
- Zhang, B., et al., Mesenchymal Stem Cells Secrete Immunologically Active Exosomes. Stem Cells and Development, 2014. 23(11): p. 1233-1244.
- Zhang, B., et al., MSC-sEV Treatment Polarizes Pro-Fibrotic M2 Macrophages without Exacerbating Liver Fibrosis in NASH. International Journal of Molecular Sciences, 2023. 24(9): p. 8092.
- Teo, K.Y.W., et al., Mesenchymal Stromal Cell Exosomes Mediate M2-like Macrophage Polarization through CD73/Ecto-5′-Nucleotidase Activity. Pharmaceutics, 2023. 15(5): p. 1489.

Round 2
Reviewer 1 Report
Comments and Suggestions for Authors
Thank you for the response.
Reviewer 2 Report
Comments and Suggestions for Authors
The authors have succesfully addressed the majority of my concerns.